# The Association between Sarcopenic Obesity and DXA-Derived Visceral Adipose Tissue (VAT) in Adults

**DOI:** 10.3390/nu16111645

**Published:** 2024-05-27

**Authors:** Antonino De Lorenzo, Leila Itani, Marwan El Ghoch, Giulia Frank, Gemma Lou De Santis, Paola Gualtieri, Laura Di Renzo

**Affiliations:** 1Section of Clinical Nutrition and Nutrigenomic, Department of Biomedicine and Prevention, University of Rome Tor Vergata, Via Montpellier 1, 00133 Rome, Italy; delorenzo@uniroma2.it (A.D.L.); gemmaloudesantis@gmail.com (G.L.D.S.); paola.gualtieri@uniroma2.it (P.G.); laura.di.renzo@uniroma2.it (L.D.R.); 2Department of Nutrition and Dietetics, Faculty of Health Sciences, Beirut Arab University, Riad El Solh, Beirut 11072809, Lebanon; l.itani@bau.edu.lb; 3Center for the Study of Metabolism, Body Composition and Lifestyle, Department of Biomedical, Metabolic and Neural Sciences, University of Modena and Reggio Emilia, 41125 Modena, Italy; 4PhD School of Applied Medical-Surgical Sciences, University of Rome Tor Vergata, Via Montpellier 1, 00133 Rome, Italy; giulia.frank@ymail.com

**Keywords:** body composition, central fat, overweight, obesity, sarcopenia

## Abstract

Many people with overweight and obesity are affected by sarcopenia, which is represented by a phenotype known as sarcopenic obesity (SO), characterized by excessive body fat (BF), combined with reduced muscle mass and strength. In this population, it is vital to identify the factors associated with SO. With this aim in mind, we investigated the association between visceral adipose tissue (VAT) mass and SO in patients with overweight or obesity in a nutritional setting. A total of 256 participants (23.8% female) with overweight or obesity were involved and completed a body composition assessment, including VAT mass, using dual-energy X-ray absorptiometry (DXA). The sample was initially categorized according to whether the individual had the SO phenotype; they were then classified according to their VAT mass into three tertiles (lowest, medium, and highest). Among the 256 participants, who had a median body mass index (BMI) of 29.3 (interquartile range (IQR): 27.0–32.4) kg/m^2^ and a median age of 51.0 (IQR: 47.0–54.0) years, 32.4% were identified as having SO, and they displayed a higher median VAT mass (517.0 (IQR: 384.5–677.0) vs. 790.0 (IQR: 654.0–1007.0) g; *p* < 0.05). The logistic regression model that accounted for age, sex and BMI revealed that a higher VAT mass increases the risk of SO (odds ratio (OR) = 1.003; 95% confidence interval (CI): 1.001–1.004; *p* < 0.05). In conclusion, VAT mass appears to be an independent factor associated with SO in people with overweight or obesity. However, due to the cross-sectional design, no information regarding any causality between higher VAT mass and SO can be provided. Additional longitudinal research in this direction should therefore be conducted.

## 1. Introduction

In 2022, the World Health Organization (WHO) released a report on the prevalence of obesity in Europe, which stated that approximately 60% of people in this region have either overweight or obesity [1]. Within this population, in the past two decades, one issue has gained particular attention to the extent that it is currently considered a hot topic: sarcopenic obesity (SO) [2], which is represented by a phenotype that combines increased body fat and reduced muscle mass and strength [3]. It has been well established through extensive research that SO is a prevalent condition in individuals with overweight or obesity, especially those within a nutritional setting who are seeking or already undergoing weight management programs [4]. In those, SO was found to be associated with several medical (i.e., noncommunicable diseases, etc.) [5] as well as psychosocial comorbidities (i.e., depression, impaired health-related quality of life (HRQoL)) [6] and an increased rate of mortality [7] when compared to those with overweight or obesity but without SO. Moreover, SO appears to be associated with a decreased resting energy expenditure [4], a more sedentary lifestyle [4], and reduced physical fitness [4]. In addition, SO also seems to negatively impact weight management outcomes, involving higher attrition rates (i.e., dropout) [8] and major obstacles in long-term weight loss maintenance among individuals with SO when compared to those without SO [9]. For these reasons, the routine screening and identification of SO during weight management programs have been emphasized and recommended. The identification of factors associated with SO in individuals affected by overweight or obesity, especially in weight management settings, has therefore become vital and has important clinical implications [4]. In fact, a recent literature review that focused on identifying all the potential factors reported the top five (insulin resistance, dyslipidemia, exercise training, inflammation, and hypertension) related to SO development [10].

In this direction, a new body composition in people with overweight or obesity has attracted interest, which is represented the by fat mass located in central regions, known as visceral adipose tissue (VAT) mass [11]. A link between higher VAT mass and an increased risk of several cardiometabolic diseases has been reported [12,13,14,15,16,17,18] and it seems to be driven by the proinflammatory role of VAT mass through the secretion of certain cytokines and adipokines [19]. In this context, recent investigations have confirmed the high levels of certain proinflammatory markers and cytokines (i.e., tumor necrosis factor alpha (TNF-α), interleukin-6 (IL-6), etc.) in people with sarcopenia, suggesting that VAT mass may act directly or indirectly in SO physiopathology [20,21,22]. However, since measuring the VAT mass in an accurate way requires advanced techniques such as a computed tomography (CT) scan or magnetic resonance imaging (MRI) [23], which are not always easily available in primary care settings, few studies have assessed the association between SO and VAT mass, meaning that more research is still needed on the topic [24]. We hypothesize that a link exists between higher VAT mass and a significant probability of having SO. Therefore, and based on these considerations, our paper aims to examine the connection between VAT mass and the SO phenotype in patients with overweight or obesity within a clinical nutritional setting.

## 2. Materials and Methods

### 2.1. Participants and Design of the Study

The current research involved a cross-sectional observational investigation. The sample of this study was composed of participants who derived from a larger cohort of patients who were previously enrolled in the Section of Clinical Nutrition located at the Department of Biomedicine and Prevention, at the University of Rome “Tor Vergata”, in Italy, in the period from 1 March 2023 to the end of March 2024. A total of 256 participants who fulfilled the following inclusion criteria were included in the investigation: (i) having an age more than or equal to 20 years; (ii) being affected by either overweight or obesity, based on the WHO body mass index (BMI) cut-off, which is 25.0 kg/m^2^ or more; and (iii) underwent a body composition assessment using a dual-energy X-ray absorptiometry (DXA) scan including VAT estimation. On the other hand, the exclusion criteria were an age < 20 years, having an under- or normal weight status according to the WHO BMI cut-offs (<25 kg/m^2^), being pregnant or in the lactation period at the time of enrollment, receiving medications or treatments known to have an impact on body weight or body composition compartments or suffering from medical or psychiatric conditions that may determine a reduction in body weight (e.g., tumors, major depression).

A post-hoc power analysis was conducted based on the results of the 7.36 times higher risk and a power of 1.000 was obtained (PASS version 12.0) [25]. For the entirety of its duration, the investigation completely adhered to the Declaration of Helsinki and approval was obtained from the Ethics Committee of the Calabria Region Center Area Section (IRB number: 146 17/05/2018). All the patients’ data were treated as anonymous, as indicated by the European and Italian privacy regulations and laws. Moreover, all the participants gave their informed written consent.

### 2.2. Body Weight and Height

The participants’ body weight was measured with the patients in light clothes and without shoes using an electronic weighing scale (SECA 2730-ASTRA, Hamburg, Germany) and their height was measured by a standard wall stadiometer. The BMI was calculated by dividing the body weight expressed in kilograms by the square of the height expressed in meters [26], as reported below:BMI = Body weight ÷ (height)^2^

### 2.3. Body Composition

All the patients underwent a complete body composition assessment by means of a DXA (Primus, X-ray densitometer; software version 1.2.2, Osteosys Co., Ltd., Guro-gu, Seoul, Republic of Korea) fan beam scanner, which assessed both the whole and segmental compartments in terms of fat and lean mass. Before the assessment, each patient received full detailed and standardized instructions relating to the entire procedure of the test [27]. SO was defined as the appendicular lean mass (ALM) (kg), divided by body weight (kg) (ALM/weight), as indicated in previous studies [28,29]. Another investigation composed of a similar population to ours was used to define SO (<2 standard deviations (SD)) with the sex-specific mean of the healthy Italian group, which had an age range between 20 and 39 years [30], considered as a reference. Accordingly, the cut-offs determined from the reference group and then utilized to define SO were < 0.2827 for males and 0.2347 for females [31]. The VAT mass was estimated with Primus software version 1.2.2, Osteosys Co™, Guro-gu, Seoul, Republic of Korea [32]. To facilitate assessing the associations, VAT mass was categorized into three categories based on tertiles: lowest, middle, and highest.

### 2.4. Statistical Analysis

The normality of the data was assessed using the Kolmogorov–Smirnov test. With the rejected normality assumption, the descriptive statistics in this study are reported as medians and interquartile ranges (IQRs), and frequencies and proportions for continuous and categorical variables, respectively. The Mann–Whitney U test was applied for the comparison between the groups and the chi-squared test for the independence of categorical variables. The Pearson’s correlation coefficient was employed to examine the association and illustrated by scatterplots. Simple and multivariate logistic regression models were utilized to estimate the odds for SO with increasing VAT mass or across tertiles of VAT. The regression models were adjusted for age, sex, and BMI as potential confounders. Age and sex were included in the model based on the literature, while BMI and VAT were significant predictors in the simple logistic regression models (*p* < 0.001). An assumption for logistic regression regarding the multicollinearity of independent variables was tested by correlation analysis (correlation < 0.8). The assumption of the linearity of the relationship between the logit and independent variables was assessed with a scatter plot and visual inspection. All analyses were conducted with SPSS version 26 [33] and NCSS version 12 [25]. The level of significance for all tests was set at *p* < 0.05.

## 3. Results

A total of 256 participants with overweight (56.8%) or obesity (43.2%) and a median BMI of 29.3 (IQR: 27.0–32.4) kg/m^2^ were included in the current analysis. The sample included 195 (76.2%) males and 61 (23.8%) females with a median age of 51.0 (IQR: 47.0–54.0) years. Overall, 32.4% of the sample had SO, with a similar prevalence among males (32.8%) and females (31.1%). The participants with SO had a significantly higher BMI (32.9 (IQR: 29.6–37.0) kg/m^2^) compared to those without (28.0 (IQR: 26.7–30.2) kg/m^2^). When comparing body composition, those with SO had a significantly lower ALM (25.7 (IQR: 21.3–28.8) kg vs. 27.7 (IQR:23.6–29.8) kg) and ALM per kg body weight (0.31 (IQR: 0.29–0.33 vs. 0.26 (IQR:0.23–0.27)) and higher VAT mass (517.0 (IQR: 384.5–677.0) g vs. 790.0 (IQR: 654.0–1007.0)g) (Figure 1) (Table 1). Among those with SO, 60.0% were in the highest tertile of VAT mass when compared to those without SO (20%) (Table 1) (Figure 2).

The correlation between VAT mass (g) and ALM per kg body weight is depicted in Figure 3, which illustrates a significantly negative association (ρ = −0.392, *p* < 0.001). The effect size of the absolute VAT mass on SO was further estimated by logistic regression analysis, where each one-gram increase in VAT mass was shown to increase the risk of SO by 0.3% (odds ratio (OR) = 1.003; 95% confidence interval (CI) = 1.001–1.004, *p* < 0.05) after adjusting for age, sex, and BMI (Table 2). Comparing the risk of SO across tertiles of VAT mass revealed that it increased fivefold (OR = 5.319; 95%CI = 1.940–14.587, *p* < 0.05) for those in the second tertile of VAT mass and sevenfold (OR = 7.365; 95%CI = 2.666–20.342, *p* < 0.05) for those in the third tertile compared to those in the lowest tertile (Table 3), when adjusting for age, sex, and BMI.

## 4. Discussion

### 4.1. Findings and Concordance with Previous Studies

In this study, which used the DXA technique and aimed to assess the association between VAT mass and the SO phenotype in a clinical sample of adult patients of both sex with overweight or obesity in a nutritional setting, we found that individuals classified as having SO displayed a significantly higher VAT mass when compared to their counterparts without SO, and the former (i.e., those with SO) were more likely to fall within the highest VAT tertile, whereas the latter (i.e., those without SO) were classified as more prone to belong to the lowest VAT mass tertile. Moreover, after adjustment by potential confounding variables, namely, age, sex, and BMI, there was still a significant link between VAT mass and SO, as VAT mass remained the only independent factor to predict SO, and even a very small increase in VAT mass (≈10 gr) appears to significantly increase the risk of having SO by almost 3%. In general, our findings are in line with the previous literature that identified specific compartments of body composition, rather than solely gross corpulence (i.e., body weight or BMI), associated with the SO phenotype, such as the total BF [34]. However, the total BF remains a generic measurement since it does not distinguish subcutaneous adiposity, which is a protective compartment for health outcomes, from visceral adiposity, which is known to be associated with cardiometabolic disturbances [12,13,14,15,16].

More specifically, our findings are in accordance with the very few previous works on the topic, such as, for instance, Kim et al., who conducted a longitudinal study in a Korean population with a mean age similar to ours (≈50 years), although with a mean BMI that fell within the normal weight status, and investigated the relationship between VAT storage, as measured by CT scan, and muscle mass. They showed that the baseline visceral adiposity (i.e., VAT mass) predicts future muscle mass loss in the immediately following years and concluded that their findings may provide novel insights into SO [35]. The mechanisms underlying the association between VAT mass and the SO phenotype are complex, but due to the cross-sectional nature of our research, we were not in a position to clarify them. However, we speculate that a link between VAT mass and SO exists, and the former (i.e., VAT mass) seems to have a role that goes beyond acting as a simple fat deposit site, and is a highly secretory organ, especially in people with overweight and obesity [36]. More specifically, VAT secretes several adipokines and cytokines that determine a local and systematic, chronic, low-grade inflammation that induces lipolysis, hyperlipidemia and, following that, fat redistribution with an ex novo accumulation in the form of visceral fat, determining a further increase in the VAT mass itself [36]. However, at a certain extent the VAT mass starts to “spillover” fat to other organs of the body such as the skeletal muscle known as ectopic fat infiltration (intermuscular adipose tissue, IMAT) [37]. Both phenomena mentioned above—namely, the (i) chronic low-grade systematic inflammation and (ii) muscle ectopic fat infiltration that demonstrated VAT mass had a central role—can potentially determine a mitochondrial dysfunction contributing to the SO physiopathology [38].

### 4.2. Study Strengths and Limitations

This study has certain strengths. To the best of our knowledge, it is one of the few works to investigate the link between VAT mass and SO in similar populations to the one in this analysis (i.e., those with overweight and obesity) [35]. However, our paper also has several limitations: First, the cross-sectional design can be considered a limitation, since the association found does not provide information regarding any causality between higher VAT mass and SO, and additional longitudinal research should therefore be conducted [39]. Second, the fact that we studied a specific population (i.e., patients living in the Italy, undergoing an outpatient weight management program, etc.) means that our findings cannot be generalized to others (i.e., patients from other ethnicities) and our investigation thus lacks external validity in other settings [40]. Third, specific DXA software (Primus software version 1.2.2) was employed to estimate the VAT mass, which, although an acceptable and validated technique, especially in individuals with obesity [41], is not considered a gold standard measure for this purpose [23]. Fourth, while defining SO, we relied on a definition that considers only the muscle mass but not strength, which is another important component in defining SO, according to the recent consensus statement released by the European Society for Clinical Nutrition and Metabolism (ESPEN) and the European Association for the Study of Obesity (EASO) [42]. Finally, we were not in a position to understand the exact nature and direction of the interaction between VAT mass and the SO phenotype, or, in other words, ascertain whether VAT determines SO or vice versa.

### 4.3. Clinical Implications and New Directions for Future Research

Our findings confirm the importance of the assessment of VAT mass in individuals with overweight or obesity due to its strong connection with SO. However, we are aware that sophisticated imaging procedures for this purpose are expensive and are not always available, and therefore cannot be implemented in standard clinical settings. In this area, efforts are needed to address some of these shortcomings. For instance, it would be useful to develop simple predictive equations that would enable an easy estimation of VAT mass by anthropometry [43,44]. However, before further research is conducted, firstly, these anthropometrical algorithms should be developed for different specific populations based on ethnicity, sex, and other factors. Secondly, the identification of precise cut-off points for VAT mass that can discriminate individuals with SO from those without is certainly essential, because based on current knowledge, the values of VAT mass are absolute and difficult to interpret at the clinical level. Finally, our findings also support the notion that VAT mass should be part of the definition of overweight and obesity, considering its link with SO. Finally, a better understanding of the direction of the association/relationship (cause–effect) between VAT and SO is necessary and additional analyses will be required to unravel the underlying mechanisms (i.e., inflammatory markers, etc.) of this form of ectopic fat accumulation and its contribution (i.e., if it exists) to the risk of SO.

## 5. Conclusions

Our study shows that visceral adiposity was strongly associated with SO in patients affected by overweight or obesity, as assessed by BMI according to WHO classification. This finding needs to be replicated in future research that overcomes the limitations mentioned above, and if confirmed, visceral adiposity could therefore be considered when defining weight status beyond BMI and when developing strategies to reduce the health burden of excess adiposity.

## Figures and Tables

**Figure 1 nutrients-16-01645-f001:**
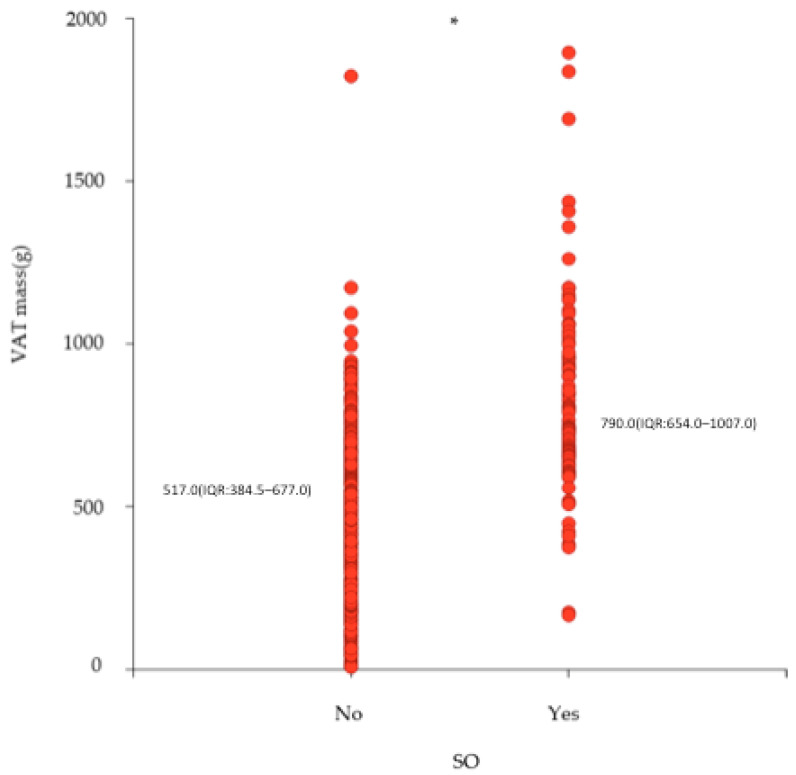
Scatterplot and median (interquartile range (IQR)) of VAT mass according to whether they are with or without SO (n = 256). VAT = visceral adipose tissue; SO = sarcopenic obesity; * *p* < 0.0001.

**Figure 2 nutrients-16-01645-f002:**
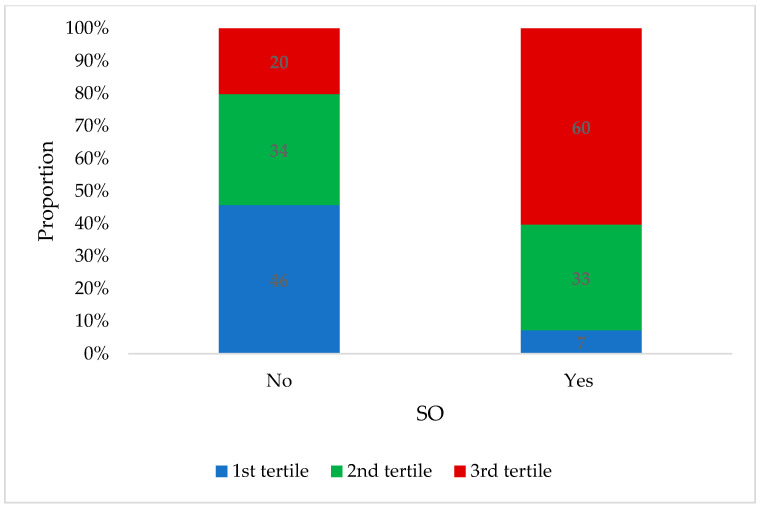
Distribution of participants across tertiles of VAT mass according to whether they are with or without SO (n = 256). VAT = visceral adipose tissue; SO = sarcopenic obesity.

**Figure 3 nutrients-16-01645-f003:**
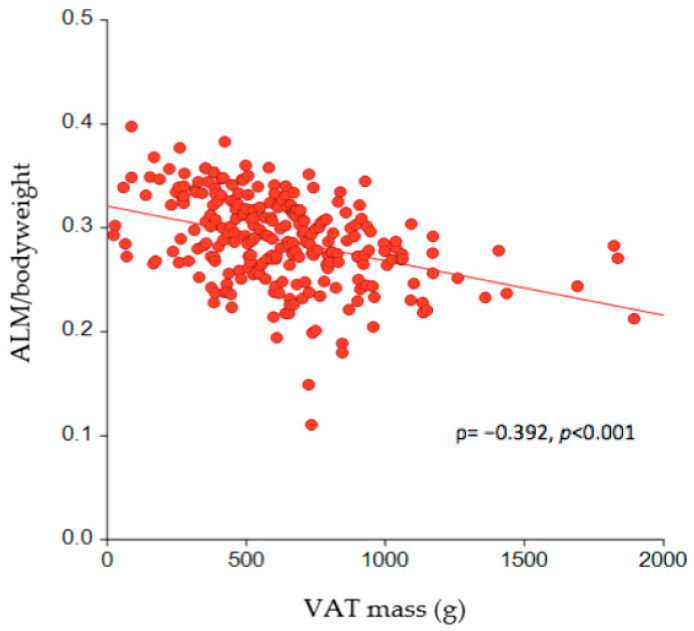
The association between VAT mass and ALM per kg of body weight in the study participants (n = 256). VAT = visceral adipose tissue; ALM = appendicular lean mass.

**Table 1 nutrients-16-01645-t001:** Demographic and body composition parameters of the study participants categorized based on whether they have SO ^§^.

	Total N = 256	Non SO N = 173	SO N = 83	Significance ^¥^
Age (years)	51.0 (47.0–54.0)	50.0 (47.0–54.0)	52.0 (48.0–55.0)	*p* = 0.186
Sex				X^2^ = 0.059; *p* = 0.808
Males	195 (76.2)	131 (75.7)	64 (77.1)	
Females	61 (23.8)	42 (24.3)	19 (31.1)	
Weight (kg)	88.2 (80.5–100.8)	85.9 (78.3–93.6)	98.1 (86.9–111.4)	*p* < 0.0001
Height (m^2^)	174.0 (168.0–179.0)	174.0 (168.0–179.7)	174.0 (168.5–176.6)	*p* = 0.459
BMI (kg/m^2^)	29.3 (27.0–32.4)	28.0 (26.7–30.2)	32.9 (29.6–37.0)	*p* < 0.0001
				X^2^ = 49.1; *p* < 0.0001
Overweight	146 (56.8)	124 (71.7)	21 (25.3)	
Obesity	111 (43.2)	49 (28.3)	62 (74.7)	
ALM (kg)	27.0 (22.8–29.4)	27.7 (23.6–29.8)	25.7 (21.3–28.8)	*p* = 0.004
ALM/body weight ratio	0.29 (0.26–0.32)	0.31 (0.29–0.33)	0.26 (0.23–0.27)	*p* < 0.0001
ALM (%)	28.7 (26.4–31.9)	30.8 (28.7–33.1)	25.6 (23.0–27.3)	*p* < 0.0001
VAT mass (g)	608.0 (433.8–793.0)	517.0 (384.5–677.0)	790.0 (654.0–1007.0)	*p* < 0.0001
				X^2^ = 52.0; *p* < 0.0001
1st tertile	85 (33.2)	79 (45.7)	6 (7.2)	
2nd tertile	86 (33.6)	59 (34.1)	27 (32.5)	
3rd tertile	85 (33.2)	35 (20.2)	50 (60.2)	

^§^ Values are medians (interquartile range (IQR)) for continuous variables and frequency (%) for categorical variables. ^¥^ Significance is reported as *p*-values for Mann–Whitney U test and *p*-value for chi-squared test.

**Table 2 nutrients-16-01645-t002:** Logistic regression analysis presenting the odds for SO with increasing VAT mass (n = 256).

	Simple Regression	Multivariate Regression
	OR (95%CI)	
Age (years)	1.002 (0.972–1.034)	1.019 (0.980–1.059)
Sex		
Males	1.00	1.00
Females	0.926 (0.499–1.719)	1.349 (0.625–2.911)
BMI (kg/m^2^)	1.412 (1.283–1.554)	1.308 (1.179–1.451)
VAT (g)	1.004 (1.003–1.006)	1.003 (1.001–1.004)

SO = sarcopenic obesity; VAT = visceral adipose tissue expressed in grams; BMI = body mass index; OR = odds ratio; CI = confidence interval.

**Table 3 nutrients-16-01645-t003:** Logistic regression analysis presenting the odds for SO with increasing tertiles of VAT mass (n = 256).

	Simple Regression	Multivariate Regression
	OR (95%CI)	
Age (years)	1.002 (0.972–1.034)	1.017 (0.977–1.058)
Sex		
Males	1.000	1.00
Females	0.926 (0.499–1.719)	1.244 (0.571–2.712)
BMI (kg/m^2^)	1.412 (1.283–1.554)	1.354 (1.217–1.506)
VAT (g)		
1st tertile	1.000	1.000
2nd tertile	6.025 (2.338–15.529)	5.319 (1.940–14.587)
3rd tertile	18.810 (7.380–47.943)	7.365 (2.666–20.342)

SO = sarcopenic obesity; VAT = visceral adipose tissue expressed in grams; BMI = body mass index; OR = odds ratio; CI = confidence interval.

## Data Availability

The original contributions presented in this study are included in the article; further inquiries can be directed to the corresponding author.

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
