# Peer review of "The Association between Sarcopenic Obesity and DXA-Derived Visceral Adipose Tissue (VAT) in Adults"

_nutrients, 2024, doi:10.3390/nu16111645_

Round 1

Reviewer 1 Report

Comments and Suggestions for Authors

It has been well established through extensive research that sarcopenic obesity is a prevalent condition in individuals with overweight or obesity. The novelty of the study is to assess the association between sarcopenic obesity and visceral adipose tissue mass in overweight or obese patients in a clinical nutritional setting. The article is well organized; it includes Introduction, Materials and Methods, Results, Discussion and Conclusions. The results are clearly presented in 3 tables and 3 graphs. Conclusions are clearly presented.

I recommend reducing self-references to less than 5% of total references.

Author Response

Reviewer #1

It has been well established through extensive research that sarcopenic obesity is a prevalent condition in individuals with overweight or obesity. The novelty of the study is to assess the association between sarcopenic obesity and visceral adipose tissue mass in overweight or obese patients in a clinical nutritional setting. The article is well organized; it includes Introduction, Materials and Methods, Results, Discussion and Conclusions. The results are clearly presented in 3 tables and 3 graphs. Conclusions are clearly presented.

I recommend reducing self-references to less than 5% of total references.

Response: We thank the reviewer for the very kind comments. Regards the self-citation, ours is only 11%, with then permitted rate, especially where all the included self-citation were needed references.

Reviewer 2 Report

Comments and Suggestions for Authors

1) The authors should report their method for assessing the normality assumption.

2) Are all the data normally distributed? The statistical analysis section only mentions the Student's t-test and Pearson's correlation.

3) The authors should provide more information regarding the development of the multivariate logistic regression (MLR) models. How were the models constructed? How were the assumptions of the MLR evaluated?

4) I am not sufficiently confident that all the data are normally distributed. Therefore, some continuous variables in Table 1 should be reported using the median and interquartile range (IQR). Please verify this.

5) In the discussion section, the authors compare their findings solely with those of one other study. Are there no other studies addressing a similar research question? Is there not a single study with contrary findings?

Comments on the Quality of English Language

Minor editing of English language required

Author Response

Reviewer #2

1) The authors should report their method for assessing the normality assumption. Response: Done as requested, now in the Method section under the statistical analysis subsection we specified the method for assessing the normality assumption (Page 3; paragraph 4).

2) Are all the data normally distributed? The statistical analysis section only mentions the Student's t-test and Pearson's correlation. Response: Based on Kolmogrov-sminov test, the data was not normally distributed and corresponding tests now are indicated in the statistical analysis subsection (Page 3; paragraph 4).

3) The authors should provide more information regarding the development of the multivariate logistic regression (MLR) models. How were the models constructed? How were the assumptions of the MLR evaluated? Response: All the requested information regarding the multivariate logistic regression and the models construction are now clearly reported (Page 3; paragraph 4).

4) I am not sufficiently confident that all the data are normally distributed. Therefore, some continuous variables in Table 1 should be reported using the median and interquartile range (IQR). Please verify this. Response: Done as requested in the Abstract (Page 1 in the abstract) and Results section (Page 4, paragraph 1) as well as in Table 1 and Figure 1.

5) In the discussion section, the authors compare their findings solely with those of one other study. Are there no other studies addressing a similar research question? Is there not a single study with contrary findings? Response: To the best of our knowledge, our study is one of the few works to investigate the link between VAT and SO in similar populations (i.e., with overweight and obesity). Now this is clearly mentioned in the Discussion section (Page 8; paragraph 1).

Reviewer 3 Report

Comments and Suggestions for Authors

The manuscript titled " The Association Between Sarcopenic Obesity and DXA-Derived Visceral Adipose Tissue (VAT) in Adults." is submitted to the journal for potential publication in the "Nutritional Obesity" section.

The objective of this study is to investigate the relationship between visceral adipose tissue (VAT) mass and sarcopenic obesity (SO) among adult patients who are overweight or obese within a nutritional context.

Comments:

The title accurately reflects the content of the study.

The abstract aligns well with the main content of the study, although the conclusion is somewhat concise, considering the cross-sectional observational design where causality is not demonstrable due to the modest odds ratio (OR) value.

The keywords appropriately capture the essence of the study.

 The introduction effectively introduces the topic of visceral adipose tissue (VAT) mass and sarcopenic obesity (SO) using relevant literature. However, it should be supplemented with information on factors potentially involved in sarcopenic obesity (SO) in adults based on current knowledge. This condition arises from multiple factors, and the introduction should reflect this complexity. Additionally, the hypothesis should precede the presentation of the objectives.

Materials and Methods: A cross-sectional observational design was utilized, which primarily describes rather than analyzes, and its outcomes should be interpreted cautiously. The manuscript should specify if a sample size calculation was performed for the stated objective, and the participation rate of individuals meeting the inclusion criteria should be reported.

Results: The 256 participants were classified based on the presence or absence of SO. The results are clearly presented, and the tables and figures effectively convey the study's ideas. In Tables 2 and 3, where multivariate regression is applied, the manuscript should detail which variables were included for adjustment.

 In the discussion, the obtained results are analyzed; however, caution should be exercised in their interpretation due to the study's design, sample size, and the complex nature of factors influencing SO, as acknowledged by the authors in the limitations section. These limitations should be considered when drawing conclusions.

Author Response

Reviewer 3#

The manuscript titled " The Association Between Sarcopenic Obesity and DXA-Derived Visceral Adipose Tissue (VAT) in Adults." is submitted to the journal for potential publication in the "Nutritional Obesity" section. The objective of this study is to investigate the relationship between visceral adipose tissue (VAT) mass and sarcopenic obesity (SO) among adult patients who are overweight or obese within a nutritional context.

Comments:

The title accurately reflects the content of the study. Response: We thank the reviewer for the comment.

The abstract aligns well with the main content of the study, although the conclusion is somewhat concise, considering the cross-sectional observational design where causality is not demonstrable due to the modest odds ratio (OR) value. Response: We thank the reviewer for the comment; now a statement on regard has been added to the Abstract to make conclusions less concise (Page 1 in the abstract).

The keywords appropriately capture the essence of the study. Response: We thank the reviewer for the comment.

The introduction effectively introduces the topic of visceral adipose tissue (VAT) mass and sarcopenic obesity (SO) using relevant literature. However, it should be supplemented with information on factors potentially involved in sarcopenic obesity (SO) in adults based on current knowledge. This condition arises from multiple factors, and the introduction should reflect this complexity. Additionally, the hypothesis should precede the presentation of the objectives. Response: We thank the reviewer for the valuable comment. Now we added information on factors potentially involved in SO in adults based on current knowledge aligned with suitable reference. Moreover, as suggested we moved the hypothesis to precede the objective of the study (Page 2; paragraph 1).

Materials and Methods: A cross-sectional observational design was utilized, which primarily describes rather than analyzes, and its outcomes should be interpreted cautiously. The manuscript should specify if a sample size calculation was performed for the stated objective, and the participation rate of individuals meeting the inclusion criteria should be reported. Response: A post hoc power analysis was conducted and reported in text (Page 3; paragraph 1).

Results: The 256 participants were classified based on the presence or absence of SO. The results are clearly presented, and the tables and figures effectively convey the study's ideas. In Tables 2 and 3, where multivariate regression is applied, the manuscript should detail which variables were included for adjustment. Response: Now in the entire manuscript, the variables that were included for adjustment have been clearly mentioned.

In the discussion, the obtained results are analyzed; however, caution should be exercised in their interpretation due to the study's design, sample size, and the complex nature of factors influencing SO, as acknowledged by the authors in the limitations section. These limitations should be considered when drawing conclusions. Response: As suggested, now the conclusions are stated with caution taking into account the limitations of our study (Page 8; paragraph 3).

Round 2

Reviewer 3 Report

Comments and Suggestions for Authors

I have carefully reviewed the new version of the manuscript " The Association Between Sarcopenic Obesity and DXA-Derived Visceral Adipose Tissue (VAT) in Adults "( nutrients-3026971), as well as the authors' response to the reviewers' suggestions for improving the comprehensibility of their work.

It is evident that the authors have addressed the requested information and acknowledged the limitations arising from the study design and sample size employed.

Minor comments: In line 97, it is mentioned that 256 participants met the inclusion criteria. Please specify what these inclusion criteria are. If a sample size calculation was not performed for this study, it should be indicated that the sample is one of convenience.

Author Response

Reviewer 3#

I have carefully reviewed the new version of the manuscript " The Association Between Sarcopenic Obesity and DXA-Derived Visceral Adipose Tissue (VAT) in Adults "(nutrients-3026971), as well as the authors' response to the reviewers' suggestions for improving the comprehensibility of their work. It is evident that the authors have addressed the requested information and acknowledged the limitations arising from the study design and sample size employed.

Response: Thankful for the reviewer comment.

Minor comments: In line 97, it is mentioned that 256 participants met the inclusion criteria. Please specify what these inclusion criteria are.

Response: The inclusion and exclusion criteria are now clearly mentioned.

If a sample size calculation was not performed for this study, it should be indicated that the sample is one of convenience.

Response: A post hoc power analysis has been conducted, and clearly mentioned in the Method section (Page 3, paragraph 1).